# Proteomics Analysis Reveals that Warburg Effect along with Modification in Lipid Metabolism Improves In Vitro Embryo Development under Low Oxygen

**DOI:** 10.3390/ijms21061996

**Published:** 2020-03-14

**Authors:** Qaisar Shahzad, Liping Pu, Armughan Ahmed Wadood, Muhammad Waqas, Long Xie, Chandra Shekhar Pareek, Huiyan Xu, Xianwei Liang, Yangqing Lu

**Affiliations:** 1State Key Laboratory for Conservation and Utilization of Subtropical Agro-bioresources, College of Animal Science and Technology, Guangxi University, Nanning, Guangxi 530000, China; raoqaisarshahzad@gmail.com (Q.S.); 18775352516@163.com (L.P.); armughanwadood@gmail.com (A.A.W.); waqas_sk@yahoo.com (M.W.); mrchildren77582580@163.com (L.X.);; 2Institute of Veterinary Medicine, Faculty of Biological and Veterinary Sciences, Nicolaus Copernicus University, 87-100 Torun, Poland; pareekcs@umk.pl; 3Division of Functional genomics in biological and biomedical research, Centre for Modern Interdisciplinary Technologies, Nicolaus Copernicus University, 87-100 Torun, Poland; 4Guangxi Key Laboratory of Buffalo Genetics and Breeding, Buffalo Research Institute, Chinese 10 Academy of Agriculture Science, Nanning 530001, China; liangbri@126.com

**Keywords:** buffalo, embryo, oxygen, proteome, warburg effect

## Abstract

The molecular mechanism regulating embryo development under reduced oxygen tension remains elusive. This study aimed to identify the molecular mechanism impacting embryo development under low oxygen conditions. Buffalo embryos were cultured under 5% or 20% oxygen and were evaluated according to their morphological parameters related to embryo development. The protein profiles of these embryos were compared using iTRAQ-based quantitative proteomics. Physiological O_2_ (5%) significantly promoted blastocyst yield, hatching rate, embryo quality and cell count as compared to atmospheric O_2_ (20%). The embryos in the 5% O_2_ group had an improved hatching rate of cryopreserved blastocysts post-warming (*p* < 0.05). Comparative proteome profiles of hatched blastocysts cultured under 5% vs. 20% O_2_ levels identified 43 differentially expressed proteins (DEPs). Functional analysis indicated that DEPs were mainly associated with glycolysis, fatty acid degradation, inositol phosphate metabolism and terpenoid backbone synthesis. Our results suggest that embryos under physiological oxygen had greater developmental potential due to the pronounced Warburg Effect (aerobic glycolysis). Moreover, our proteomic data suggested that higher lipid degradation, an elevated cholesterol level and a higher unsaturated to saturated fatty acid ratio might be involved in the better cryo-survival ability reported in embryos cultured under low oxygen. These data provide new information on the early embryo protein repertoire and general molecular mechanisms of embryo development under varying oxygen levels.

## 1. Introduction

Buffalo is a major source of milk, meat and other dairy products, which boosts the economy of various developing countries. The importance of this species is evidenced by the fact that it fulfils almost 13% of the world’s milk demand [1]. Due to the increasing world population, there is a dire need and a huge demand to enhance livestock productivity. Therefore, assisted reproductive techniques (ART) are being applied to improve livestock genetics [2]. In particular, in vitro embryo production (IVEP) has been used to strengthen knowledge regarding oocyte and embryonic growth [3], producing low-cost embryos for basic research, the propagation of elite female genetics, in combination with sex sorted sperm technology for generation of sexed embryos [4], and the application of embryo technologies like nuclear transfer and transgenesis.

The IVEP system has significantly improved over the past few decades, resulting in higher blastocyst yield, pregnancy development and birth rate in various mammalian species [5,6,7,8]. Besides the improvement, it is still considered an inefficient technique because only 5% of matured oocytes lead to live birth [9]. The major issue impeding embryonic development in vitro is the oxidative stress. Indeed, in vitro, higher atmospheric oxygen tension (20%) increased the generation of ROS compared to that in the female reproductive tract (2%–8%) [10]. ROS production is further accelerated through the metabolism of spermatozoa, oocytes and the embryo as well as during the thawing of media. These ROS can diffuse through the cell membrane and alter cellular molecules like proteins, lipids and nucleic acids. There are several consequences of ROS diffusion into cells, such as mitochondrial damage, ATP depletion, DNA damage, membrane disruption due to lipid peroxidation, embryo developmental block and apoptosis [11].

During in vivo fertilization, the embryo is protected by oxygen scavengers present in oviductal and follicular fluid, but in vitro these are absent. There are several strategies to reduce oxidative stress during the process of IVEP, including the reduction in oxygen to a physiological level [12] and the addition of antioxidants in culture media [13]. Discrepancies are reported regarding oxygen tension during IVEP. Several studies showed that there is no effect of oxygen on embryonic development [14,15], while others reported higher blastocyst rate or lower apoptosis in lowered oxygen tension (5%–7%) in several species [16,17,18]. It is therefore important to elucidate whether regulating oxygen tension can be a means to improve IVEP.

The efficiency of IVEP can be enhanced through understanding the molecular mechanism of embryo development. Many approaches have greatly expanded our understanding of the molecular mechanism associated with embryo development, such as transcriptomics, epigenomics, single cell and microcell sequencing [19,20,21,22,23,24,25]. However, proteomics might provide us with a direct insight into the molecular mechanism governing embryo development, as proteins are the executors of most of the developmental programs. Moreover, proteomic analysis has been conducted to comprehend the mechanism of oocyte and embryo development in various species including buffalo [26,27], cows [28,29,30], mice [31,32,33,34,35,36] and pigs [37,38,39,40]. However, there is no study available determining the role of oxygen tension on in vitro embryo development at the proteome level. In the present study, we have cultured the buffalo embryos under atmospheric oxygen (20%) or physiological oxygen (5%) and evaluated the morphological parameters of embryo development. Simultaneously, we have analyzed these embryos for alterations in the proteome at the hatched blastocyst stage by using an iTRAQ-based mass spectrometry technique [41].

## 2. Results

### 2.1. Evaluation of the Effect of Oxygen Tension (5% vs. 20%) on Morphological Parameters of Embryo Development

#### 2.1.1. Cleavage, Blastocyst, Hatching Rate and Developmental Kinetics

The cleavage and blastocyst rate are the number of cleaved embryos and blastocysts normalized to the total number of oocytes (inseminated), respectively. The hatching rate is the number of hatched blastocysts normalized to the total number of blastocysts. The analysis of morphological parameters of embryo production indicated that 5% O_2_ had no effect on cleavage rate (50.46 ± 1.76 vs 51.1 ± 2.53) as compared to atmospheric oxygen. However, 5% significantly (*p* < 0.05) improved blastocyst rate (34.08 ± 2.40 vs 24.28 ± 0.51) as well as increased hatching rate (83.67 ± 2.14 vs 68.7 ± 1.38) as compared to 20% O_2_ (Figure 1).

The developmental kinetics of embryos were evaluated at day 5, 6, 7 and 8 of culture by assessing developmental stages of embryos. The physiological O_2_ group had a more significant number of embryos at advanced stages on the particular day of development as compared to the atmospheric O_2_ group (Figure 2). 

#### 2.1.2. Cell Count

The cell count was determined by staining the hatched blastocysts with Hoechst 33342. These morphological findings of embryo development were further reinforced by the cell count evaluation, as shown in Figure 3, which revealed that embryos cultured under 5% O_2_ had significantly (*p* < 0.05) higher cell counts as compared to the embryos developed under 20% O_2_ (Table 1).

#### 2.1.3. Embryo Quality Scoring

The embryo quality scoring system was based on the definitions presented in the manual of the European Society of Human Reproduction and Embryology (see Materials and Methods section). Embryos were scored from score 1 (less advanced embryos) to score 6 (hatched blastocysts). The results based on the embryo quality scoring suggest that significantly (*p* < 0.05) better quality of embryos are produced under low oxygen as compared to atmospheric oxygen culture (Figure 4). The physiological oxygen group had a lower proportion of low score embryos and a higher proportion of the high score embryos in comparison with the 20% O_2_ group.

#### 2.1.4. Cryo-Survival Potential of Vitrified Blastocysts

The cryo-survival ability of vitrified blastocysts was determined by evaluating hatching rate post-warming (Table 2). The results show that reduced oxygen embryo culture plays an essential role in enhancing the cryo-survival ability of the blastocysts, as revealed by significantly higher hatching rate (post-warming) under low oxygen embryo culture in comparison to atmospheric oxygen culture (82.67% ± 4.44% vs. 60.69% ± 1.80%).

### 2.2. Proteomic Analysis of Embryos Cultured under 5% or 20% Oxygen Tension

The precision of proteomic quantification was evaluated by Pearson’s correlation coefficient and coefficient of variance. The stability of the proteomic analysis was confirmed by a significant correlation between the three technical replicates (Figure 5A). The correlation coefficient was more than 0.60, which represents a good correlation among the replicates of each group. Figure 5B shows the percentages of proteins with a mean CV < 10%, 10% ≤ CV < 20%, and CV ≥ 20%. Overall, more than 90% of proteins had less than 10% coefficient of variance. Proteins with >20% coefficient of variance were less than 1% in the dataset, confirming the reliability of the analysis. A total of 954 proteins were identified in embryos cultured at 20% or 5% oxygen, and 566 proteins were found to be shared among all the replicates (Figure 6A). The differential expression analysis (log_2_FC [$\frac{5\%}{20\%}$] > 20%, *p < 0.05*) showed 25 upregulated and 18 downregulated proteins (Figure 5 and Figure 6, Appendix A). Out of 43 differentially expressed proteins (DEPs), five were uncharacterized, which have been omitted from the heatmap (Figure 6B). The DEPs were mainly associated with metabolic pathways such as aerobic glycolysis, lipid degradation, terpenoid backbone synthesis and inositol phosphate metabolism. 

#### 2.2.1. Gene Ontology Analysis of Differentially Expressed Proteins

The differentially expressed proteins were further classified by gene ontology annotation based on three categories: biological process, cellular component, and molecular function. The GO analysis was derived from the online database (http://bioinformatics.sdstate.edu/go/, ShinyGO v0.60, 18-09.2019). After biological processes GO analysis of proteins upregulated in 5% O2 embryos, we found that the biological processes include metabolic process, cellular component organization and biological regulation (Figure 7A). The molecular function of these proteins was important in structural molecule activity, catalytic activity and antioxidant activity. The results of the cellular component demonstrated that these proteins were mainly associated with membrane, organelle and protein-containing complexes. The GO analysis of downregulated proteins was conducted and analyzed the biological process (Figure 7B). We found that it includes cellular process, metabolic process and developmental process proteins. The molecular functions mainly include binding and catalytic activity. The results of the cellular component include cell part, organelle part and protein-containing complex.

#### 2.2.2. KEGG Analysis of Differentially Expressed Proteins

The identified differentially expressed proteins were analyzed based on the Kyoto Encyclopedia of Genes and Genomes (KEGG) database. Our results identify that upregulated proteins in 5% O2 embryos were mainly enriched in glycolysis, pyruvate metabolism, fatty acid degradation, terpenoid backbone synthesis and inositol phosphate metabolism (Figure 8). The downregulated proteins in 5% O2 embryos were mainly enriched in phagosome activity (Figure 8).

#### 2.2.3. Proteome Profile Validation by Real Time Quantitative PCR

The proteomics data were validated by real time quantitative PCR. The four genes regulating aerobic glycolysis were selected for the validation of the results. The gene involved in glucose transport (GLUT3, also known as SLC3A2), a key regulator of glycolysis, phosphofruktokinsae-1 (PFK1), glycolysis intermediate, glyceraldehyde phosphate dehydrogenase (GAPDH) and the gene responsible for the conversion of pyruvate to lactate (LDHA) were selected for RT-qPCR. The qPCR analysis revealed significant differences in mRNA expression level between 5% oxygen and 20% Oxygen among all four up-regulated proteins and thus validated the findings of proteomic analysis (Figure 9).

#### 2.2.4. Fluorescent intensity of Lipids

The intensity of lipid droplets was measured using Nile red staining. The analysis showed that fluorescent intensity was higher (*p* < 0.05) in blastocysts cultured under atmospheric oxygen (Figure 10A). The representative micrographs of hatched blastocysts cultured under atmospheric and physiological oxygen are shown in Figure 10B,C, respectively. 

## 3. Discussion

Oxygen regulation is crucial to embryonic development. During the preimplantation development in vivo, the embryo experiences a low-oxygen microenvironment, typically ranging from 2% to 8% [10,42,43,44,45]. This oxygen tension contrasts with that present in the atmosphere (20%). Simultaneously, perusal of the literature has suggested a growing consensus on the beneficial effects of lower oxygen on in vitro embryo development [46,47,48]. However, a few studies have shown no improvement in embryo developmental potential under reduced oxygen tension [14,15], signifying the importance of evaluating the influence of oxygen on embryos of individual species. Furthermore, the molecular mechanism of oxygen mediating embryo development is not yet clear. Therefore, we have evaluated the embryo development under reduced and atmospheric oxygen levels and determined the molecular mechanism of embryo development under different oxygen levels by using LC-MS/MS based proteomics approach.

### 3.1. Physiological Oxygen Improves Morphological Parameters of Embryo Development

During the embryo development in vivo, the requirement of oxygen varies according to the stage of embryonic development [49]. Such precise control of oxygen level is not possible or difficult to maintain in vitro. Due to this reason, it would be more feasible to find a specific oxygen level at which maximum embryo development may be obtained. It is well established that an elevated oxygen level leads to reactive oxygen species production, which in turn generates oxidative stress and, consequently, cell death occurs [50,51]. The present study has shown that physiological oxygen does not affect the cleavage of embryos. However, it increases the proportion of embryos developing to the blastocyst and hatched blastocyst stage. There are studies that have shown improved embryo development under low oxygen [12]. However, a few studies have suggested that a higher O_2_ level is more suitable for embryo production in bovines [52]. There are also studies which have reported that varying oxygen level does not affect the in vitro embryo development [15,53]. The variable results could be attributed to the use of different types of culture medium. We have shown that embryos had greater developmental kinetics under 5% O_2_ and they have higher cell count, suggesting that physiological O_2_ is more suitable for buffalo embryo production. Moreover, blastocyst produced under low oxygen had a higher cryo-survival ability, as demonstrated by higher hatching rate post-warming. 

### 3.2. Embryos Developed under Low Oxygen Showed Pronounced Warburg Effect

The present study has shown that embryos under low oxygen exhibit, in terms of proteomic profile, the metabolic adaptation, recognized as the Warburg Effect (Figure 11). The characteristic feature of the Warburg Effect is the increased glucose uptake and fermentation of glucose to lactate, supports rapid cell growth [54]. The glucose uptake by the cell is facilitated by GLUT3 protein, also known as Solute Carrier Family 2 Member 3 (SLC2A3), which is a potential high-affinity glucose carrier. During embryo development, GLUT3 is specific to the morula and blastocyst stages [55]. The increased expression of GLUT3 suggests a higher uptake of glucose by embryos incubated in a low oxygen environment, which is a signature of more viable embryos [56,57]. 

One of the hallmarks of the Warburg Effect is that pyruvate is directed away from the tri-carboxylic acid cycle and metabolized to lactate, resulting in a buildup of glycolytic intermediates [58]. The present study demonstrated the accumulation of enzymes involved in six chemical reactions of glycolysis (Figure 9) in embryos cultured under 5% O_2_. Although ATP production from Warburg Effect is less efficient as compared to oxidative phosphorylation, it efficiently meets the demand for the generation of macromolecules, such as nucleic acid, in order to facilitate cell proliferation [58,59,60]. The elevated levels of glycolysis intermediates are proposed to increase R5P (Ribose 5-phosphate) and NADPH production, which are required for nucleic acid production and maintaining cellular redox status. 

Pyruvate, the end product of glycolysis, may enter the TCA cycle or convert to lactate. The conversion of the pyruvate to lactate is catalyzed by LDHA, which is supposed to maintain the NAD^+^ level that supports elevated glycolysis [61]. The embryos under low oxygen showed increased expression of LDHA, which is the key feature of the Warburg Effect. The real time quantitative PCR validated the findings of the proteomics data, as the protein involved in glucose uptake (GLUT3), glycolysis intermediates (PFK1, GAPDH) and protein involved in lactate production (LDHA) showed upregulation under 5% O_2_. Taken together, embryos cultured under physiological oxygen levels showed pronounced features of the Warburg Effect, which may be associated with increased developmental competence of embryos under low oxygen.

### 3.3. Modification in the Lipid Composition might be Associated with Increased Cryo-Survival Ability of Embryos Cultured under Low Oxygen 

Buffalo embryos have higher lipid content, which hampers the success of cryopreservation [62]. The present study has shown that buffalo embryos cultured under physiological oxygen had a higher cryo-survival ability. The proteomic analysis demonstrated the upregulation of proteins involved in the degradation of lipids and cholesterol biosynthesis in embryos cultured under physiological oxygen. Simultaneously, the fluorescent intensity of lipids, an indicator of the density of lipid droplets [63], was low in embryos cultured under low oxygen. Here, we suggested that the better cryo-survival ability of the embryos under 5% O2 may be related to the changes in lipid metabolism, ending in a lower content of lipid droplets.

Generally, the utilization of the IVEP is important for genetic improvement and selection [64]. Embryos generated through IVEP should be immediately transferred to recipients or cryopreserved. Management of recipient herds for immediate transfers is costly and time consuming. Therefore, embryo cryopreservation is the preferred method, as fewer management procedures are required by using this technique. However, cryo-survival rate of in vitro produced embryos is significantly lower as compared to in vivo embryos [65]. Eventually, the conception rate of cryopreserved in vitro embryos is considered too low for commercial use. 

Attempts to improve embryo cryopreservation are based mainly into two major approaches: modification of the cryopreservation technique or cells themselves. All mammalian cell membranes consist of the lipid bilayer that mainly contains phospholipids. Each phospholipid has two fatty acid tails that can differ in length and saturation level. The properties of fatty acid tails are essential for strengthening the membrane for cryopreservation. Cryopreservation can damage membrane integrity, thereby causing membrane chilling injuries [66,67]. Cryoresistance of the cells may be improved through making their membranes more fluidic [68]. Membrane fluidity is influenced by the level of unsaturation in phospholipids and the amount of cholesterol present in the membrane. 

In the present study, the proteomic analysis showed elevated expression of proteins involved in cholesterol synthesis (HMGCS1 and HMGCR) in embryos cultured under low oxygen (Figure 9). 3-hydroxy-3-methylglutaryl-coa synthase 1 (HMGCS1) is an enzyme that condenses acetyl-CoA with acetoacetyl-CoA to form HMG-CoA, which is the substrate for HMG-CoA reductase (HMGCR). HMGCR is a rate limiting enzyme in the cholesterol biosynthetic pathway [69]. The increased expression of HMGCS1 and HMGCR suggests that embryos cultured under low oxygen may have a higher cholesterol level that could be the reason for their better cryo-survival ability. Increasing the cholesterol level of the membrane of bovine cells has been characterized to improve cryotolerance [68]. The KEGG analysis of differentially expressed proteins from the present study suggested that low oxygen embryos had increased expression of fatty acid degradation proteins (ACAT2 and ACSL4). These findings are supported by the low lipid fluorescent intensity in embryos under physiological oxygen. The fluorescent intensity is an indicator of density of lipid droplets [63]. The higher lipid level is attributed to the poor cryo-survival ability of embryos [62]. The proteomic and fluorescent intensity analysis implies that the better cryo-survival ability of embryos under low oxygen may be due to lower lipid levels. Acetyl-coenzyme A acetyltransferase 2 (ACAT2) is an enzyme which catalyzes the reaction and converts two acetyl Co-A into acetoacetyl Co-A [70]. Acyl-CoA synthetase long chain family member 4 (ACSL4) is an enzyme that utilizes free long chain fatty acids and converts them into fatty acyl-CoA esters, and thereby play a key role in lipid biosynthesis and fatty acid degradation. This enzyme preferentially utilizes arachidonic acid as a substrate [71]. As arachidonic acid is a polyunsaturated fatty acid, we propose that increased expression of ACSL4 protein may have enhanced the level of unsaturation in the phospholipid bilayer of the membrane, thereby improving the cryo-survival ability of embryos cultured under low oxygen. However, we recommend that these hypotheses should be confirmed by using lipidomics approach.

## 4. Materials and Methods 

All the chemicals and reagents were obtained from Sigma-Aldrich (St. Louis, MO, USA), unless otherwise noted. No animals were used for this work. All studies were conducted on slaughterhouse-derived materials. The schematic representation of the experimental design is briefly described in Figure 12. 

### 4.1. Morphological Parameters of Embryo Development under Different Oxygen Tension (5% vs. 20%)

#### 4.1.1. In Vitro Oocyte Maturation, Fertilization and Culture

Buffalo ovaries were collected from a local abattoir in Nanning, China, and immediately (within 2 h) transported to the laboratory. Cumulus oocyte complexes (COCs) were aspirated from follicles (2–8 mm) using 18 G needles connected to 10-mL disposable syringes containing 2-mL collection medium (Hepes-buffered TCM-199 supplemented with 3% newborn serum) and subsequently washed 3 times in the same medium. Good quality COCs (oocyte with intact and compact cumulus cells) were selected and matured at 38.5 °C under 5% CO_2_ air atmosphere for 22–24 h in maturation medium (TCM-199 supplemented with 10% fetal bovine serum, 100 mM cysteamine, 0.5 μg/ mL FSH, 1 μg/mL E_2_ and 5 μg/mL LH). Fertilization was then performed in 20 µL of fertilization media (Tyrode’s medium with 0.6% BSA, 2.5 mM caffeine sodium benzoate, and 20 mg/mL heparin) for 6–8 h in 5% CO_2_ at 38.5 °C. At the end of insemination, cumulus cells and sperms were separated by gentle pipetting and presumptive zygotes were washed twice in Hepes-buffered TCM 199 medium (supplemented with 3% newborn bovine serum) and three times in embryo culture medium, BO-IVC (Falmouth, UK). The presumptive zygotes were randomly divided into two groups and cultured in a standard incubator (5% CO_2_ and 20% O_2_) or in a low-oxygen chamber (5% CO_2_ and 5% O2) at 38.5 °C for eight days. The experiment was conducted in 3 replicates, and a total of 576 and 567 embryos were cultured in the 20% and 5% O_2_ groups, respectively. The blastocyst and hatching rate were assessed at day 7, and development rates were recorded at days 5, 6, 7 and 8 post-inseminations.

#### 4.1.2. Cell Staining

Cell staining was performed on hatched blastocysts from both (*n* = 15 each) the groups on day 7. After washing three times in Dulbecco’s Phosphate Buffer Saline (DPBS), embryos were stained with Hoechst (33342, 1 mg/mL) for 25 min in the dark at room temperature. The slides were sealed with glycerin and cells were counted under an epifluorescence microscope under ×200 and ×400 magnifications (Figure 3 in the results section).

#### 4.1.3. Embryo Quality Scoring 

The embryo quality scoring system was established for the buffalo embryos according to the definitions presented in the manual of the European Society of Human Reproduction and Embryology. Briefly, blastocysts stage embryos were specified with different scores (1–6) based on the degree of expansion. The blastocysts with less than 50% blastocoel cavity were allotted score 1 (Figure 13A); score 2 blastocysts had half or more than half blastocoel cavity (Figure 13B); score 3 blastocysts were those in which the blastocoel completely filled the embryo (Figure 13C); blastocoel was greater than the original volume of the embryo in score 4 blastocysts (Figure 13D); score 5 blastocysts or hatching blastocysts (Figure 13E) in which trophoblast cells were herniating through natural breach of zona pellucida (ZP); and score 6 blastocysts or hatched blastocysts were those in which the blastocysts had completely escaped through a natural breach in ZP (Figure 13F). 

#### 4.1.4. Vitrification, Thawing and Hatching Rate of Blastocysts

All equilibration, vitrification and thawing solutions were prepared in 25 mM HEPES-buffered TCM199 supplemented with 10% FBS. Solutions were brought to room temperature before use. Vitrification was performed according to the method reported by Yang et al. [31]. Briefly, one end of a 0.25-mL straw was cut out with a sharp scalpel to form an open pole. Blastocysts (non-hatched) were equilibrated in equilibration solution for 5 min, then transferred into a vitrification solution A (10% ethylene glycol + 10% DMSO) for 30 sec and another 30 sec in vitrification solution B (20% ethylene glycol + 20% DMSO + 0.5M sucrose) and immediately placed on the tip of the open pole, and rapidly plunged into liquid nitrogen. Straws were stored in liquid nitrogen for at least 24 h before thawing. Thawing was performed in thawing solution (0.5 M sucrose) for 5 min, and then embryos were placed in equilibration medium for 5 min. Subsequently, blastocysts were cultured for 72 h and assessed for hatching rate.

#### 4.1.5. Statistical Analysis

The data obtained for cleavage, blastocyst and hatching rate were statistically analyzed by using Fisher’s exact test. The data obtained for embryo quality scoring were analyzed by using the chi-square test. The data for cell count were normalized by arcsine transformation and tested by Wilcoxon test. All the statistical analysis was performed using GraphPad Prism 7 software

### 4.2. Proteomic Analysis of Embryos Cultured under 5% or 20% Oxygen Tension

#### 4.2.1. Cell Lysis

The proteomic analysis was performed by extracting proteins from the hatched blastocysts (*n* = 6 blastocyst per replicate) cultured under varying oxygen levels (5% or 20%) in 3 biological replicates. The cells were lysed in 20 µL lysis buffer (5 mM EDTA, 5 mM Egtazic acid, 10 mM NaOH, 1× Protease Inhibitor Cocktail, 10 mM HEPES (pH 8.5), 10 mM dithiothreitol, 1%SDS) and homogenized by ice-cooled sonification for 10 min. Then, the precipitation of proteins was done by adding 4 µL processed beads [72] and 20 µL of TFE (trifluoroethanol) and performing the sonification of each tube in the ice bath for 15 min. After this, 0.75 µL of 0.1% formic acid was added to each tube, and then lysate was incubated at 95 °C for 5 min and placed on ice for 30 sec. Samples were incubated at 45 °C for 30 min and then, 5 µL 400 mM iodoacetamide (IAA) were added to each tube and incubated at 24 °C for 30 min. To stop the reaction, 5 µL of 200 mM DTT (dithiothreitol) was added to each tube.

#### 4.2.2. Protein Purification

For protein purification, 1% formic acid (5 µL) was added to 10 µL of the lysate (in order to acidify the solution) and then 15 µL acetonitrile was added to the solution. The mixture was incubated at room temperature for 8 min, and the tube was then placed at a magnetic stand for 2 min at room temperature. The supernatant was discarded, and the pellet was washed with 70% ethanol (200 µL). The pellet was washed twice with 70% ethanol. Then, the supernatant was removed, and acetonitrile (180 µL) was added into the pellet and shifted to the magnetic stand. The pellet solution was further incubated for 15 min dried at room temperature for 30 sec. The peptide digestion was performed by adding 5 uL digestion solution (5:1 of 50 mM HEPES and trypsin) and incubating at 37 °C for 14 h.

#### 4.2.3. Peptide Recovery and Preparation

The 5 µL of digested peptides and 195 µL acetonitrile were mixed in a tube and incubated for 8 min. The PCR tube was placed at the magnetic stand and incubated again at room temperature for 2 min. The supernatant was discarded and 180 µL acetonitrile was added to the magnetic stand and incubated for 15 sec. After the removal of the supernatant, 2% DMSO was added to reconstitute the magnetic bead and sonicated for 1 min to improve the recovery rate. The tube was centrifuged briefly (2 sec) and placed on the magnetic stand. The supernatant was separated in a new tube containing 1 µL of 1% formic acid. 

#### 4.2.4. iTRAQ Labelling

For iTRAQ labelling, peptides were reconstituted in 12 µL of 500 mM triethylammonium bicarbonate (TEAB) by sonification for 5 min. Then, isopropanol (150 µL) was added to dissolve 0.8 mg of iTRAQ tag and placed for 5 min. After mixing, 15 µL iTRAQ tag was added to each tube of peptide and incubated the tube at room temperature for 2 h. Samples from 20% and 5% oxygen group were labelled as 126 and 127, respectively. The peptide mixtures were then pooled and dried by vacuum centrifugation.

#### 4.2.5. Mass Spectrometry 

The sample was reconstituted with 10 µL of mobile phase A (99.9% water and 0.1% formic acid); sonicated in a water bath for 5 min and centrifuged at 12,000 g for 5 min, and slowly added to the sample bottle with a pipette (to avoid air bubbles). The mobile phases used in the LC-MS elution: mobile phase A (98% water, 2% acetonitrile, and 0.1% formic acid); mobile phase B (98% acetonitrile, 2% water, and 0.1% formic acid). For iTRAQ-labeled samples, the loading amount was 3 µL, and the elution was performed at a rate of 300 nL/min. The maximum pressure (600 bar) was applied, and the gradient changes over 145 min were observed (buffer B is at 0%–15% for 5 min; 100 min at 15%–27%, 20 min at 27%–40%, 20 min at 40%–98%). Data were acquired on qExactive MS (Thermo Scientific) using a data-dependent method. The first 12 peaks were selected after higher-energy collisional dissociation (HCD) fragmentation for tandem-MS/MS (MS2) analysis. Scans covered a mass range of 350–1800 with a resolution of 60,000, HCD fragmentation. The normalized collision energy was set at 33.

#### 4.2.6. Database Search Settings

The protein database for *Bos Taurus* (UP000009136, 24078 sequences, published March 2017) was downloaded from Uniport database. Then, data were imported to Proteome Discoverer software v1.3 and the database search parameters were set: we selected iTRAQ8plex (+304.205 Da) for the quantification method, trypsin full for the digestion method, and set precursor mass tolerance to 20 ppm; fragment mass tolerance was set to 0.02 Da; dynamic modification was oxidation (+15.995 Da M); fixed modification was carbamidomethyl (+57.021 Da C); target FDR (false discovery rate) was set to 0.01, target FDR (Relaxed) was set to 0.05, and the verification method used q-Value.2.9. The proteome profile of each replicate (*n* = 3) is listed in Appendix A.

#### 4.2.7. Bioinformatics Analysis

The variations and closeness between the replicates of each group were assessed by calculating the coefficient of variance and correlation. The differentially expressed proteins (DEPs) between the two groups were identified by using a web based EdgeR package [73]. A *p*-value < 0.05 and log_2_FC [$\frac{5\%}{20\%}$] > 20% were considered to be significant. The log_2_FC value is based on m value [73]; if log2FC is 0, it shows no change, and if log2FC is 0.2, it means 20% change. The DEPs are listed in Appendix A. The cellular component, molecular functions and biological processes of DEPs were identified and subjected to GO functional classification and enrichment analysis. The DEPs were also subjected to KEGG enrichment analysis. For GO and KEGG analysis, an online website (http://bioinformatics.sdstate.edu/go/, ShinyGO v0.60, 18-09.2019) was used keeping the FDR at 0.05. 

#### 4.2.8. Real Time qPCR Analysis

Total RNA was extracted from hatched blastocysts by using the protocol as described previously [74]. Extracted RNA was then transcribed into cDNA by using the reverse transcription kit (GoScript^TM^ Reverse Transcription Mix, Random Primer Protocol, Promega). The Sybr green-based real time qPCR was performed in a thermocycler for real-time PCR (CFX96 Touch; BIO-RAD, USA). The primer pairs used for qPCR are listed in Appendix A. The experiment was performed in three replicates, with 10–12 blastocysts in each replicate. For each group of embryos, the expression levels of the target genes were normalized to the expression level of the endogenous control, ribosomal protein S15 (RPS-15). The RPS-15 is considered as one of the most stable endogenous control for buffalo embryos [75]. The Ct values for RPS 15 were almost similar between both of the groups. We determined the relative level of expression of each mRNA using the 2^−ΔΔCt^ (normalized expression ratio) method [76]. 

#### 4.2.9. Assessment of Lipid Accumulation

The lipid content was assessed by Nile Red staining in hatched blastocysts (*n* = 15, each group). The Nile red stock solution was prepared by dissolving Nile Red dye (Cat# R-72485) at 1 mg/mL in DMSO and stored in the dark. Obtained blastocysts were washed in 0.01% PBS-PVP for 1 min. Blastocysts were fixed in 4% formaldehyde solution at 4 °C for 24 h. After fixation blastocysts were washed in 0.01% PBS-PVP to remove the fixative and stained with Nile Red (10 μg/mL) for 3 h at room temperature in dark condition taking all necessary precautions. The extra stain was removed by washing with 1% PBS-PVP and stained blastocysts were observed under a fluorescence microscope using FITC filter to capture the images with a 40× objective. Image analysis and fluorescent intensity were measured using Image J software. The statistical analysis was done by Wilcoxon test by using GraphPad Prism 8 software.

## 5. Conclusions

This study provides an insight into the mechanism of embryo development under physiological and atmospheric oxygen. We conclude that reduced oxygen provides a more conducive environment for buffalo embryo culture and improves blastocyst yield, hatching rate, developmental kinetics, cell count, as well as cryo-survival ability. The proteomic analysis data showed that the more significant developmental potential of the embryos under low oxygen might be due to the pronounced Warburg Effect. Alongside this, the cryo-survival ability of the embryos cultured under low oxygen was better; this might be linked with an increase in the expression of cholesterol biosynthesis proteins, higher unsaturated to saturated fatty acids ratio and fatty acid degradation proteins. These data provide a rich resource for further studies on utilizing the Warburg effect and lipid metabolism for improvement in embryo development.

## Figures and Tables

**Figure 1 ijms-21-01996-f001:**
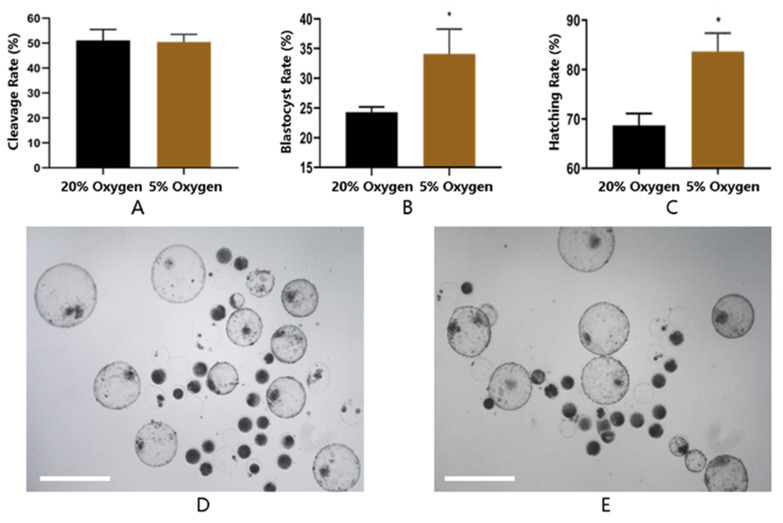
Effect of varying oxygen levels (20% vs. 5%) on embryo development. All the experiments were repeated three times. Embryos were cultured in 20% (*n* = 180) or 5% (*n* = 178) oxygen and assessed for cleavage on day 2 of culture. Oxygen did not influence the cleavage rate of embryos (**A**). Culture under 5% oxygen (*n* = 567) improved the proportionate of the embryos developing to the blastocyst stage (**B**) and hatched blastocyst stage (**C**) as compared to 20% oxygen (*n* = 576). The blastocyst and hatching rate were evaluated on day 7 of embryo development. Representative photographs of embryos cultured in 5% oxygen (**D**) and 20% oxygen (**E**). The images (1D and 1E) were captured with microscope (50i; Nikon, Japan) at 100× resolution. Asterisk (*) indicates the statistical difference (*p* < 0.05) between the groups.

**Figure 2 ijms-21-01996-f002:**
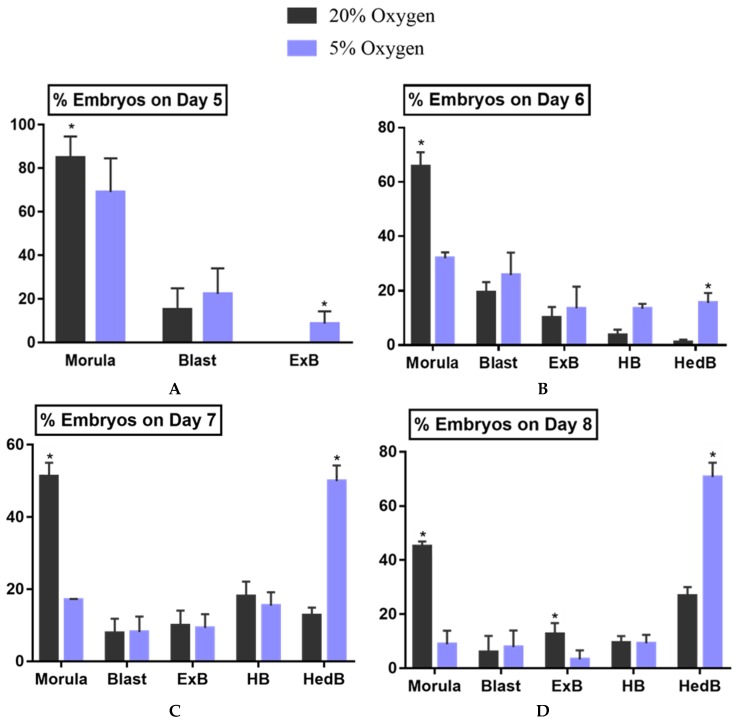
Effect of varying oxygen levels (20% vs. 5%) on developmental kinetics of buffalo embryos. All the experiments were replicated three times. The percentage of embryos at different developmental stages, morula, blastocyst (Blast), expanded blastocyst (ExB), hatching blastocyst (HB), hatched blastocyst (HedB), on day 5, 6, 7 and 8 was evaluated. There was a significantly higher rate of embryos at advanced developmental stages in 5% oxygen as compared to 20% oxygen. Fifty-nine and 53 embryos from 20% and 5% oxygen group, respectively, were evaluated on day 5 (**A**). Sixty-four embryos from each group were evaluated on day 6 (**B**). Sixty-three and 64 embryos from 20% and 5% oxygen group respectively, were evaluated on day 7 (**C**). Sixty-nine and 60 embryos from 20% and 5% oxygen group, respectively, were evaluated on day 8 (**D**). Values represent mean % ± S.E.M. Asterisk (*) indicates the statistical difference, *p* < 0.05.

**Figure 3 ijms-21-01996-f003:**
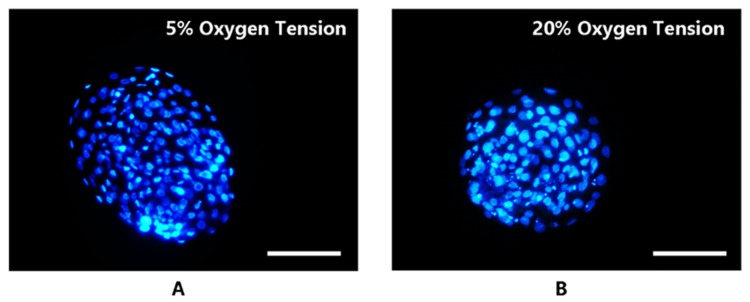
Hatched blastocysts stained with Hoechst 33342 to determine the total cell count. Blastocysts were cultured under 5% (**A**) and 20% (**B**) oxygen tension. The images were captured with microscope (50i; Nikon, Japan) at 200× resolution.

**Figure 4 ijms-21-01996-f004:**
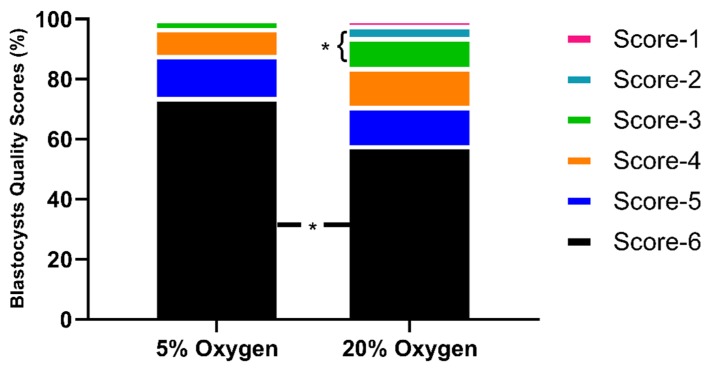
Embryo quality scoring. Physiological oxygen improved the quality of blastocysts, as indicated by lower percentage of low score blastocysts and higher proportionate of high score blastocysts. Asterisk (*) indicates the significant difference (*p* < 0.05) between the groups.

**Figure 5 ijms-21-01996-f005:**
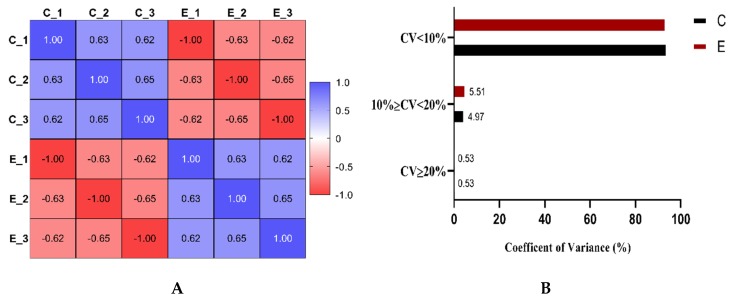
Precision of proteomic quantification. The stability of proteomics data was determined by estimating the Pearson’s correlation coefficient between the replicates of each group (C = 20% O_2_ and E = 5% O_2_). A significant correlation was found between the replicates of both groups (**A**). The reliability of the data was further supported by variance analysis, as more than 90% of the proteins from both groups had a coefficient of variance of less than 10% (**B**). For each replicate (*n* = 3), 6 hatched blastocysts from each group (*n* = 2) were included in proteomic analysis.

**Figure 6 ijms-21-01996-f006:**
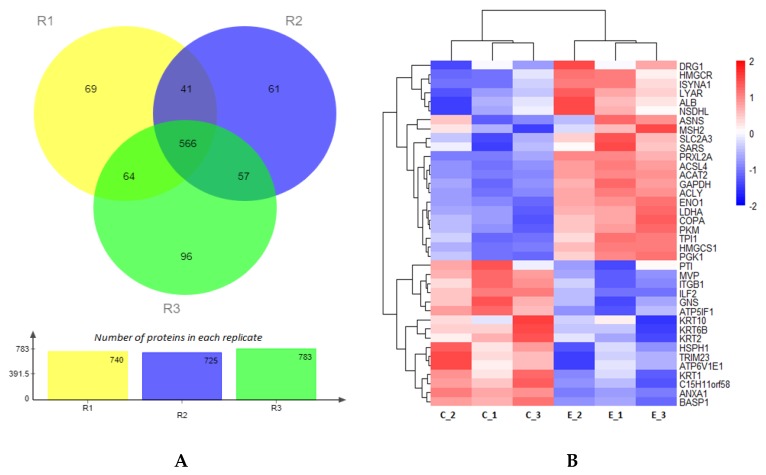
Protein profile obtained by labelled iTRAQ LC-MS/MS Analysis. Venn diagram showing the overlap of the quantified proteins within the three replicates (R1, R2 and R3). About 72.28% (566) of all the proteins (954) were found in every replicate, showing the good conformance of the samples (**A**). Heatmap presenting the distribution of differentially expressed proteins (log_2_FC [$\frac{5\%}{20\%}$] > 20%, *p* < 0.05). Pink and blue bars showing proteins that are significantly upregulated (25) and downregulated (18) in the 5% oxygen group in comparison with 20% oxygen group (**B**). The uncharacterized differentially expressed proteins have been omitted from the heatmap. C and E correspond to 20% and 5% oxygen, respectively.

**Figure 7 ijms-21-01996-f007:**
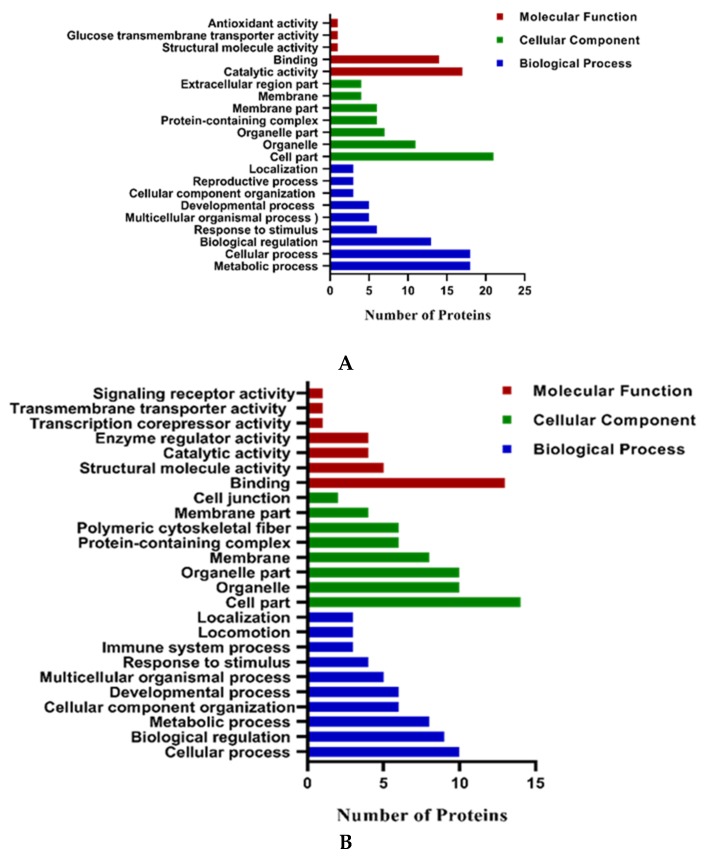
Gene ontology analysis of upregulated (**A**) and downregulated (**B**) proteins in 5% O2 embryos. Proteins were classified according to cellular localization, biological processes and molecular functions. The results are displayed as a number of proteins classified to each category.

**Figure 8 ijms-21-01996-f008:**
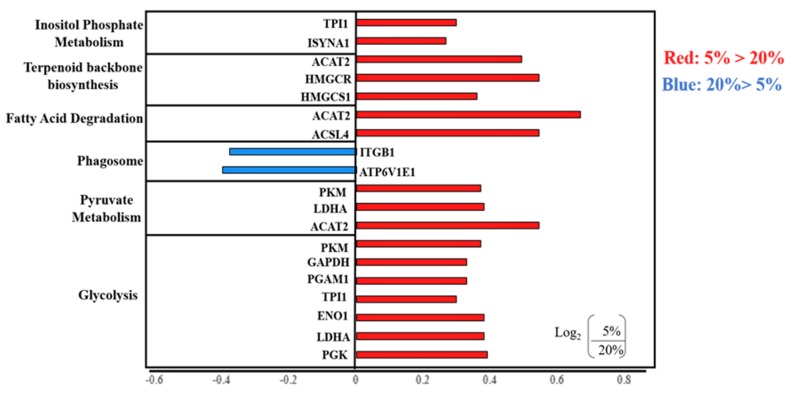
KEGG pathway analysis of differentially expressed proteins, Red colour shows upregulation and blue colour shows downregulation of protein levels in 5% oxygen group.

**Figure 9 ijms-21-01996-f009:**
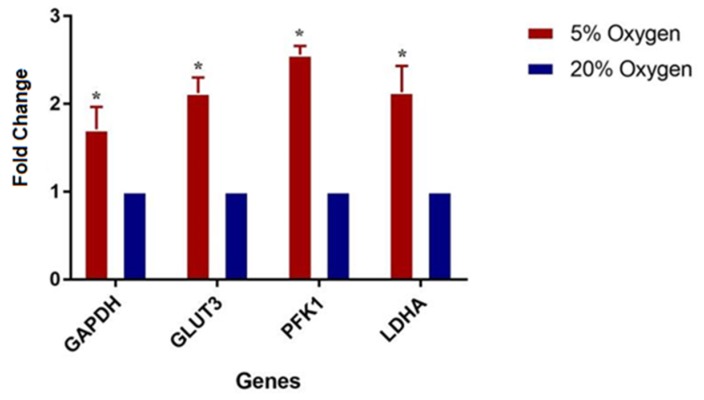
The real time qPCR analysis showing the elevated expression of GAPDH, GLUT3, PFK1 and LDHA in embryos cultured under low oxygen as compared to atmospheric oxygen. Asterisk (*) indicates the significant difference (*p* < 0.05) between the groups.

**Figure 10 ijms-21-01996-f010:**
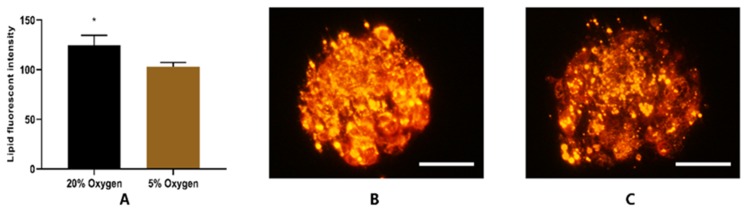
Fluorescent analysis of Lipid. Embryos cultured under atmospheric oxygen showed higher fluorescent intensity (**A**). Photomicrographs of Nile Red stained blastocysts: (**B**) 20% Oxygen and (**C**) 5% Oxygen. The images (Figure 10B,C) were captured with microscope (50i; Nikon, Japan) at 200× resolution. Asterisk (*) indicates the significant difference (*p* < 0.05) between the groups.

**Figure 11 ijms-21-01996-f011:**
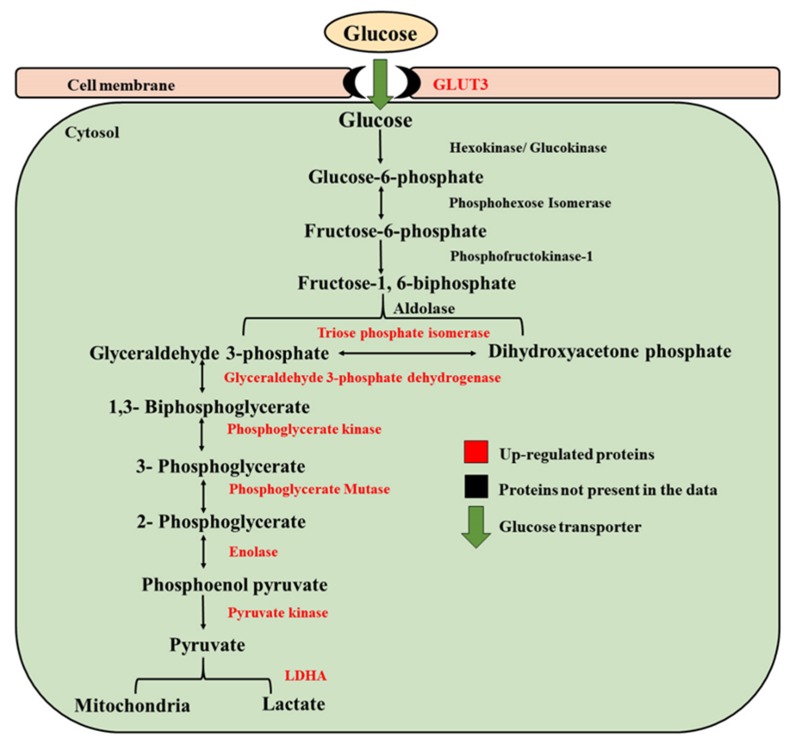
Regulation of enzymes involved in aerobic glycolysis (Warburg Effect). Red colour shows the upregulation of protein levels in 5% oxygen. Green arrow shows the transport of glucose molecule from outside of the cell to inside through cell membrane, facilitated by glucose transporter (GLUT3).

**Figure 12 ijms-21-01996-f012:**
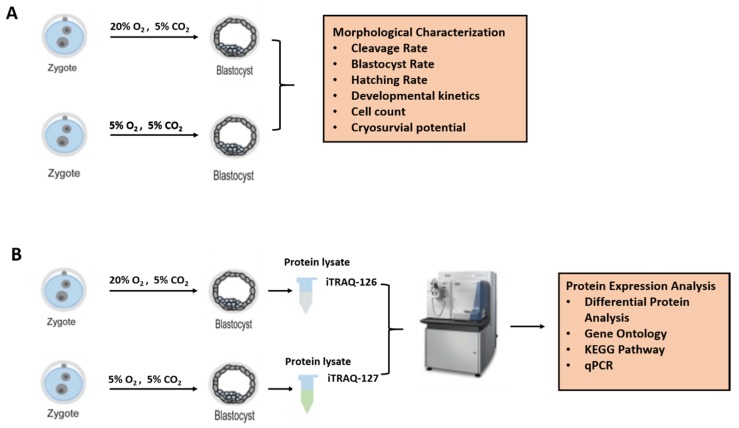
Schematic illustration of experimental design. Buffalo embryos were cultured in 5% or 20% oxygen and evaluated for morphological parameters of embryo development (**A**). Alterations in embryo proteome after culturing in different oxygen levels (5% vs 20%) were assessed by using labelling based iTRAQ quantitative proteomics approach (**B**).

**Figure 13 ijms-21-01996-f013:**
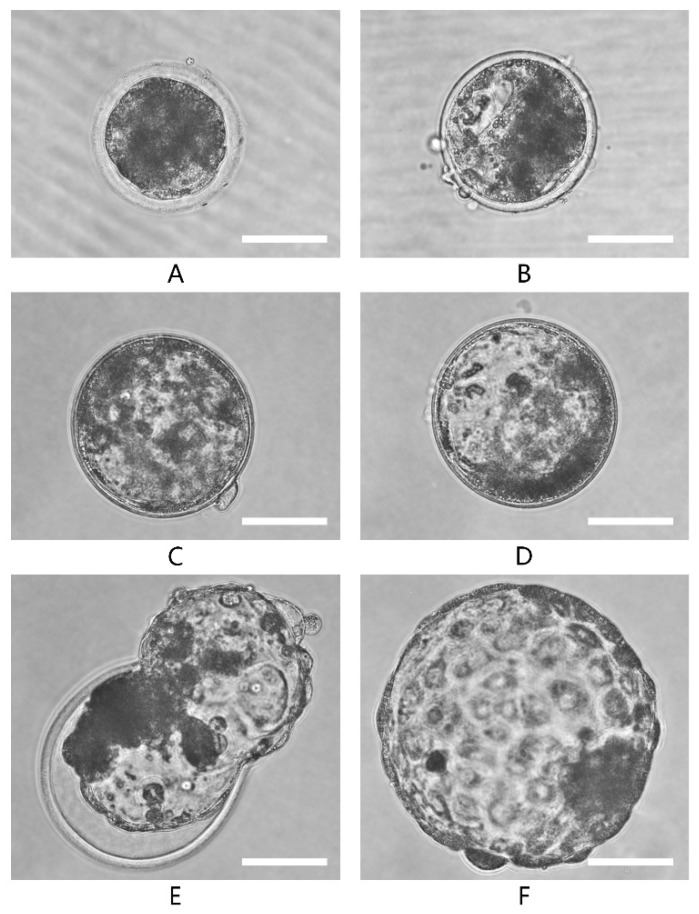
Embryo quality scoring based on expansion. Embryos were allotted scores of 1 (**A**), 2 (**B**), 3 (**C**), 4 (**D**), 5 (**E**) and 6 (**F**). The images were captured with microscope (50i; Nikon, Japan) at 200x resolution.

**Table 1 ijms-21-01996-t001:** Cell count of embryos cultured in 5% and 20% Oxygen.

Parameters.	N	Cell Count (Mean ± S.E.M)
5% Oxygen	15	151.36 ± 2.70
20% Oxygen	15	89.18 ± 2.60 *

Asterisk (*) indicates the significant difference (*p* < 0.05) between the groups.

**Table 2 ijms-21-01996-t002:** Post-warming hatching rate of the blastocysts produced under different levels of oxygen (5% and 20%).

Parameters	N	Hatching Rate (Mean ± S.E.M)
5% Oxygen	42	82.67 ± 4.44
20% Oxygen	38	60.69 ± 1.80 *

Asterisk (*) indicates the significant difference (*p* < 0.05) between the groups.

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
