# Peer review of "Proteomics Analysis Reveals that Warburg Effect along with Modification in Lipid Metabolism Improves In Vitro Embryo Development under Low Oxygen"

_ijms, 2020, doi:10.3390/ijms21061996_

Round 1

Reviewer 1 Report

General comments:

This is a potentially interesting article, providing an insight to protein expression changes in IVF buffalo blastocysts incubated at different oxygen levels during development, and I am happy that the experiments conducted do support your conclusions.

However, in its current format this manuscript shows several inadequacies. It is not easy for the reader to follow as it contains numerous typos, grammatical errors, poor figures and figure legends.

Abstract:

Line 30 – Should read ‘This study concluded..’

Line 32 – ‘Along this’ doesn’t make sense.

Introduction:

Note – here I have also noted the grammatical errors to point out how the wording should flow. I would expect if you put the text through an English grammatical check that it will highlight other errors in the manuscript. I have not noted grammatical errors in other parts of the manuscript.

Line 41 – FAO? I assume it means data from The Food and Agriculture Organization. Please expand or include a report citation.

Line 43-44 – Should read ‘Especially in vitro embryo production (IVEP) which has been used…’

Line 48 – in not into.

Line 50 – an not as

Line 51-53 – Should read ‘The major issue impeding embryonic development in vitro is the problem of oxidative stress; due to the generation of reactive oxygen species (ROS) from the higher atmospheric oxygen tension (20%) compared to that in the female reproductive tract (2-8%) [9].

Line 54 – add ‘the’ between through and metabolism.

Line 56 – remove comma after lipids and add ‘and’. Add ‘the’ between into and cell.

Line 59 - remove comma after fertilization and add ‘the’.

Line 60 – would absent be a more appropriate than deficient?

Line 61 – clarify the abbreviation ‘IVF’. Add ‘the’ between and, and addition.

Line 62 – remove ‘the’ after regarding.

Line 65 – add ‘a’ between is and need.

Line 68 - add ‘the’ between of and molecular.

Line 72 – remove the ‘s’ off proteomics. Have should be has.

Line 73 – add comma after buffalo.

Line 75 – in vitro should be in italics. Add ‘the’ between at and proteome.

Results:

Line 93 – IVC?

Line 155 – Add DEP.

Line 179 – the results do not reveal that low oxygen have lower stores of lipid or higher cholesterol. Unless you have this data change revealed to suggest (or similar).

Line 187 – GLUT3 has been shown as SLC2A3 in the previous results. nomenclature either needs changing in the heat map or explaining they’re the same protein. PFK not present in the heat map, does it have another name? If so the previous comment should be applied.

Figures

Generally, the figure legends are not stand-alone and do not describe the figures adequately.

Figure 1. – No labelling of Y axis.

Figure 2. - No labelling of Y axis. Legend very confusing.

Tables – need consistency in layout, i.e. the parameters are switched between tables.

Figure 4. Needs more description of what they show in the results text and legend.

Figure 5. Heat map needs more description in the results text and legend. Not sure if Red/Green is the best colouring option considering the most common colour blindness is red/green.

Figure 6. There are more bars than categories for A and B. Remove bracket after process. Remove part from B.

Figure 8. This pathway can be displayed so much better, the text looks stretched and is not pleasant to look at. Need to explain what process the green arrow is.

Figure 9. Y axis should state that its fold change to 20% O2.

Discussion:

Line 196 – concomitantly is not a commonly used word, consider changing.

Line 230-231 – Statement about GLUT3 implies it is only in the embryo when it is found in other tissues. Please revise wording.

Line 233 – Should read: In the present study, 8 out of 10 metabolising proteins of glycolysis appeared in the data and all 233 were up regulated, indicating an accumulation of glycolysis intermediates.

Line 235 – what is R5P?

Line 250 – ‘in vitro’ should be in italics.

Line 260 – ‘experiment’ should be study and ‘genes’ should be proteins.

Line 264 and 265 – HMGCR should be used instead of HMG-CoA reductase

Materials and Methods:

Line 299 – what is NBS?

Line 301-302 – it is not clear if the n numbers are for each replicate or are a total.

Line 326 – ‘upheld’ should be changed to brought.

Line 327 – et al should be in italics.

Line 341 – this is cell lysis not protein lysis.

Line 344 – what is the composition of the lysis buffer? If you used a commercial kit them please state this in the method.

Line 345 – what are the processed beads?

Line 348-349 – what is IAA and DTT?

Line 357 – enzymolysis of what? What is the composition of the enzymolysis solution and why is enzymolysis in italics?

Line 361 – 95% of what?

Line 366-367 – the sentence ‘Equal volume of each stage was mixed into Mixture‘ does not make sense.

Line 369 – what is TEAB?

Line 372 – ‘Next, 12 μl of A solution was reconstituted and 2 μl was added to each tube.’ Doesn’t make sense. Also at what stage were the samples digested with trypsin?

Line 373 – what is solution A’s composition?

Line 373-374 – ‘Then, 5 μl of mass spectrometry solution A was added to apply mass spectrometry. Equal amount of peptide of each stage was added and mixed.’ Does not make sense.

Line 384 – what is HCD?

Line 386-387 – ‘33 for iTRAQ peptides’ does not make sense.

Line 394 – what is FDR?

Line 406 – please state if the qPCR method was TaqMan or sybr green. Unfortunately, I was unable to open the supplementary data.

Line 413 – should be ‘ribosomal protein S15’. Also, please include the justification of using this as an endogenous control and the Ct variation between the two groups.

Conclusions:

Line 420-421 – should read ‘The proteomic analysis data showed that the greater developmental potential of the embryos under low oxygen may be due to pronounced Warburg Effect.’

Line 422 – see earlier comment about concomitantly. Change better to improved.

References:

Line 543-645 – References 1-40 are duplicated.

Author Response

Dear Reviewer

First of all, we would like to thank you for the critical reading of our manuscript and for detailed comments. We did our best to take into consideration all the comments and suggestions made by the reviewers. English of the manuscript was improved by Dr. Sehrish Akbar (PhD Biotechnology). We hope that these modifications improved our manuscript quality and readability, and will be accepted. The detailed response to your comments is in attachment.

Reviewer 2 Report

This paper provides information on the effect of either low or atmospheric oxygen on embryo development, in terms of blastocyst rates, embryo cell count, proteomic data…

Some flaws regarding statistical analysis need revision.

The paper needs to be edited for the english

Cautious on speculation needs to appear in the text

See comments in the attached file

Author Response

Dear Reviewer

First of all, we would like to thank you for critical reading of our manuscript and for detailed comments. We did our best to take into consideration all the comments and suggestions made by you. English of the manuscript was improved by Dr. Sehrish Akbar (PhD Biotechnology). We hope that these modifications improved our manuscript quality and readability, and will be accepted. The detailed response to reviewer’s comments is described below.

Round 2

Reviewer 2 Report

My recommendation is to accept the paper after minor revision (see the pdf file). (quality of presentation is now average and not low anymore)
